# Investigating the Effects of Gossypetin on Cardiovascular Function in Diet-Induced Pre-Diabetic Male Sprague Dawley Rats

**DOI:** 10.3390/ijms252212105

**Published:** 2024-11-11

**Authors:** Karishma Naidoo, Andile Khathi

**Affiliations:** Department of Human Physiology, School of Laboratory Medicine and Medical Sciences, College of Health Sciences, University of KwaZulu-Natal, Durban 4000, South Africa; karishma280199@gmail.com

**Keywords:** Gossypetin, natural flavonoid, cardiovascular, diet-induced pre-diabetes

## Abstract

Gossypetin (GTIN) is a naturally occurring flavonoid recognised for its pharmacological properties. This study examined the effects of GTIN on cardiovascular function in a diet-induced pre-diabetic rat model, which has not been previously studied. Pre-diabetes was induced using a high-fat high-carbohydrate (HFHC) diet supplemented with 15% fructose water for 20 weeks. Thereafter, the pre-diabetic animals were sub-divided into five groups (*n* = 6), where they were either orally treated with GTIN (15 mg/kg) or metformin (MET) (500 mg/kg), both in the presence and absence of dietary intervention for 12 weeks. The results demonstrated that the pre-diabetic (PD) control group exhibited significantly higher plasma triglyceride, total cholesterol, low-density lipoprotein and very low-density lipoprotein levels, along with decreased high-density lipoprotein (HDL) levels in comparison to the non-pre-diabetic (NPD) group. This was accompanied by significantly higher mean arterial pressure (MAP), body mass index (BMI), waist circumference (WC) and plasma endothelial nitric oxide (eNOS) levels in PD control. Additionally, there were increased heart malondialdehyde levels, reduced heart superoxide dismutase and glutathione peroxidase activity as well as increased plasma interleukin-6, tumour necrosis factor alpha and c-reactive protein levels present in the PD control group. Notably, both GTIN-treated groups showed significantly reduced plasma lipid levels and increased HDL, as well as decreases in MAP, BMI, WC and eNOS levels in comparison to PD control. Additionally, GTIN significantly decreased heart lipid peroxidation, enhanced antioxidant activity and decreased plasma inflammation markers. These findings may suggest that GTIN administration in both the presence and absence of dietary intervention may offer therapeutic potential in ameliorating cardiovascular disturbances associated with the PD state. However, future studies are needed to determine the physiological mechanisms by which GTIN improves cardiovascular function in the PD state.

## 1. Introduction

High-caloric diets have been linked to an increased risk of metabolic disorders, including pre-diabetes, which is a multifactorial condition influenced by genetic, environmental and lifestyle factors [1,2,3]. Pre-diabetes is an intermediate state between normal glucose tolerance [4] and overt type 2 diabetes mellitus (T2DM) [5]. The International Diabetes Federation estimated that about 298 million adults worldwide have been diagnosed with pre-diabetes, with projections showing that this number will increase to about 414 million by the year 2045 [6]. This condition is characterised by moderate hyperglycaemia, moderate insulin resistance and impaired glucose tolerance [5]. Studies have shown that many of the complications seen in T2DM begin in the pre-diabetic (PD) state [7,8]. Excess saturated fatty acids, trans fatty acids and pro-inflammatory adipokines have been shown to promote the development of insulin resistance, which is a risk factor for pre-diabetes [9,10,11,12]. Pre-diabetes has been shown to be associated with an increased risk of cardiovascular diseases (CVDs) and endothelial dysfunction [9]. This condition has been shown to promote adverse effects through multiple pathways by generating reactive oxygen species (ROS) [13]. Increased oxidative stress reduces the bioavailability of nitric oxide (NO), promoting increased blood pressure [14]. Chronic low-grade inflammation has been observed in the PD state, and this has been characterised by increased levels of pro-inflammatory markers, including tumour necrosis factor (TNF)-α, interleukin (IL)-6 and c-reactive protein (CRP) [15,16]. This has further been accompanied by increased plasma levels of triglycerides (TG), low-density lipoproteins (LDL) and decreased high-density lipoproteins (HDL), which have been shown to promote foam cell accumulation in artery walls and, subsequently, atherosclerosis [17]. 

The current strategy for the management of pre-diabetes involves both the combination of lifestyle modification and pharmacological intervention [18]. However, patients often neglect lifestyle modifications due to the financial burden of dietary changes and ease of access to pharmacological treatment [19]. Furthermore, studies have shown that individuals with lower educational backgrounds may lack knowledge of healthy dietary habits and preventive behaviours [20,21,22]. Limited nutrition education is a contributing factor to the development of pre-diabetes, as individuals with insufficient knowledge tend to make unhealthy lifestyle choices, including the consumption of high-calorie, nutrient-deficient foods that contribute to insulin resistance and metabolic dysfunction [23,24,25,26]. Additionally, a lack of understanding about the importance of dietary changes and higher prices of healthier foods may lead to an over-reliance on pharmacological treatments instead of lifestyle modifications, delaying the potential benefits of dietary intervention in reducing the risk of T2DM development [19,27,28,29]. Studies have shown that pharmacological intervention in the absence of dietary modification shows limited effectiveness in the treatment of pre-diabetes [30,31]. This highlights the need for pharmacological treatments that can effectively manage pre-diabetes and its associated complications in both the presence and absence of dietary intervention [32]. Flavonoids are natural polyphenolic compounds generated by secondary metabolism in plants [33]. They function as antioxidants and possess anti-inflammatory properties in the cardiovascular system by modulating inflammatory response pathways [34,35]. Quercetin is a well-known flavonol that has shown efficacy in alleviating hyperglycaemia, hyperlipidaemia, hypertension and oxidative stress [36,37]. Studies have shown that quercetin administration was associated with reduced CVD risk [38,39]. Interestingly, GTIN (Figure 1) is structurally similar to quercetin, with an extra hydroxyl group [40,41]. It exhibits many of the same biological activities as quercetin and has demonstrated more potent antioxidant activity [40,42,43]. The capacity of GTIN to scavenge free radicals assists in cell protection and may reduce the risk of CVDs [41,44]. Furthermore, a recent study revealed that GTIN administration significantly enhanced insulin sensitivity, improved glucose tolerance and lowered fasting blood glucose levels [45]. Despite these findings, no studies have investigated the effects of GTIN on cardiovascular complications in the PD state. Hence, this study investigated the effects of diet-induced pre-diabetes on cardiovascular complications in both the presence and absence of dietary modification. 

## 2. Results

### 2.1. Lipid Profile Markers

Lipid profile marker concentrations were measured in all experimental groups at week 32 of the treatment period (*n* = 6, per group). The results (Table 1) showed that the PD control group had significantly (*p* < 0.05) higher levels of plasma TG, total cholesterol (TC), LDL and very low-density lipoproteins (VLDL), along with lower HDL levels in comparison to the NPD group. However, PD animals receiving GTIN with a normal diet (GTIN + ND) and with a HFHC diet (GTIN + HFHC) showed a significant (*p* < 0.05) reduction in these parameters and an increase in plasma HDL levels in comparison to the PD control group. Additionally, PD animals receiving MET with a normal diet (MET + ND) and with a HFHC diet (MET + HFHC) exhibited significantly (*p* < 0.05) similar results as the GTIN-treated groups.

### 2.2. MAP Levels

The MAP levels of all the animals were determined from week 20 to week 32 of the experimental period (*n* = 6, per group). The results (Figure 2) showed that the MAP levels were significantly (*p* < 0.05) higher in the PD control group throughout the treatment period in comparison to the NPD. However, both GTIN-treated groups showed significant (*p* < 0.05) reductions to MAP levels at weeks 28 and 32 in comparison to the PD control. Additionally, both the MET-treated groups exhibited significantly (*p* < 0.05) similar results as the GTIN-treated groups at weeks 28 and 32 of the treatment period.

### 2.3. BMI and WC

The BMI and WC of all the animals were determined from week 20 to week 32 of the experimental period (*n* = 6, per group). The results (Figure 3) showed that the BMI and WC in the PD control group were significantly (*p* < 0.05) higher throughout the treatment period in comparison to the NPD group. However, both GTIN-treated groups exhibited a significant reduction in BMI and WC at week 28 and week 32 of the treatment period in comparison to the PD control. The MET + ND group showed significantly reduced BMI and WC at week 28 in comparison to the PD control. Additionally, both MET-treated groups showed significantly (*p* < 0.05) similar results at week 32 in comparison to the PD control.

### 2.4. eNOS Levels

The plasma eNOS concentrations were measured in all experimental groups at week 32 of the treatment period (*n* = 6, per group). The results (Figure 4) showed significantly (*p* < 0.05) higher eNOS levels in the PD control group in comparison to the NPD group. However, both GTIN-treated groups showed significantly (*p* < 0.05) lower eNOS levels in comparison to the PD control. The MET + ND groups exhibited significantly (*p* < 0.05) similar results in comparison to the PD control. The GTIN + HFHC and MET-HFHC groups had significantly higher eNOS levels in comparison to the NPD.

### 2.5. Lipid Peroxidation, Antioxidant and Inflammatory Markers

The concentrations of heart lipid peroxidation, antioxidants and plasma inflammatory markers were assessed in the experimental groups at week 32 of the treatment period (*n* = 6 per group). The results (Table 2) indicated that the PD control group showed significantly (*p* < 0.05) higher malondialdehyde (Bhowmik, #69) levels along with reduced heart SOD and Gpx activity in comparison to the NPD group. Furthermore, the PD control group showed significantly (*p* < 0.05) higher plasma levels of IL-6, TNF-α and CRP in comparison to the NPD group. However, both GTIN-treated groups demonstrated a significant (*p* < 0.05) reversal in the level of these markers relative to the PD control. Additionally, the MET + ND group exhibited significantly (*p* < 0.05) similar effects on heart MDA, SOD and Gpx activity in comparison to the GTIN-treated groups. Notably, both MET-treated groups significantly (*p* < 0.05) lowered plasma inflammatory markers in comparison to the PD control. 

## 3. Discussion

High-calorie diets have been shown to lead to the development of pre-diabetes by promoting insulin resistance (IR) and hyperinsulinaemia [47]. This diet further contributes to the development of obesity, oxidative stress and low-grade inflammation [47]. This combination increases the risk of developing CVDs such as hypertension, myocardial infarction and stroke [48]. Various scientific studies recommend that a combination of lifestyle and pharmacological intervention is necessary to effectively manage pre-diabetes [18,49]. However, the over-reliance on pharmacological interventions without making necessary dietary changes may hinder the benefits of dietary modifications [50,51]. Flavonoids such as quercetin have been shown to mitigate the risk of CVDs during pre-diabetes by improving endothelial function, reducing inflammation and enhancing insulin sensitivity [52]. Gossypetin is present in various plant species, including *Hibiscus sabdariffa*, *Hibiscus vitifolius*, *Gossypium herbaceum* and *Gossypium arboretum* from the Malvaceae family [53,54,55,56]. It has also been found in *Empetrum nigrum* from the Ericaceae family and *Acacia constricta* from the Fabaceae family [53,54,55,56]. Notably, the majority of studies have focused on the qualitative analysis of GTIN, with scarce studies quantifying the percentage content [57,58,59,60]. A previous study showed that of the seven compounds isolated from *Hibiscus sabdariffa* phenolic extract, GTIN exhibited the highest yield of 23% [61]. Interestingly, the phenolic content in *Hibiscus sabdariffa* ranges from 24.36 to 44.43 mg of gallic acid/100 g of dried calyces [58,62,63]. Furthermore, studies have shown that GTIN exhibits potent antioxidant and anti-inflammatory properties [41,64]. The multiple hydroxyl groups present within GTIN’s structure facilitate its ability to mitigate oxidative stress [41,64]. Despite these findings, the effects of this compound on cardiovascular function in the PD state have not been investigated. 

Under normal physiological conditions, plasma lipid levels are regulated through the assistance of lipoproteins [65]. VLDLs transport hepatic TG while LDLs primarily transport cholesterol within the bloodstream [65]. HDLs facilitate reverse cholesterol transport to the liver [66]. In individuals with normal metabolic function, the lipid profile is characterised by TC levels below 5.2 mmol/L, LDL cholesterol concentrations less than 2.6 mmol/L, HDL cholesterol levels greater than 1.1 mmol/L, TG below 1.7 mmol/L and VLDL levels less than 0.8 mmol/L [67,68,69]. In contrast, individuals with pre-diabetes have been shown to exhibit disturbed lipid profiles, which may be characterised by elevated plasma TG, TC, LDL and VLDL, along with decreased HDL levels [4,70,71,72,73]. These abnormalities may include plasma TG levels ranging from 1.7 mmol/L to 2.2 mmol/L, TC levels exceeding 5.2 mmol/L, LDL levels greater than 2.6 mmol/L, VLDL concentrations between 0.8 and 1.2 mmol/L and HDL levels below 1.0 mmol/L in comparison to healthy individuals [4,70,71,72,73]. The chronic consumption of high-calorie diets, which include high sugar, saturated fats and processed foods, leads to dyslipidaemia by inducing IR and oxidative stress [74,75,76]. These conditions stimulate increased lipolysis, which in turn promotes TG synthesis and raises plasma VLDL levels [77]. Additionally, increased ROS levels promote LDL oxidation which inhibits LDL clearance [78]. This has been associated with foam cell formation and an increased risk of developing arteriosclerosis [79]. In this study, plasma TG, TC, LDL and VLDL levels were significantly higher along with lower plasma HDL levels in the PD control in comparison to the NPD group. This study’s results corroborated with previous findings of dyslipidaemia present during pre-diabetes [31,80]. However, the GTIN + ND and GTIN + HFHC groups significantly reduced these parameters while increasing plasma HDL levels in comparison to the PD control. 

Flavonoids have been shown to reduce LDL oxidation by targeting oxidative stress and decrease TG levels by promoting fatty acid oxidation [81,82]. Quercetin promotes HDL synthesis, which helps maintain cholesterol homeostasis [83]. A previous study has shown that GTIN reduced oxidised LDL-induced foam cell formation and intracellular lipid accumulation [84]. Similarly, in our study, GTIN may have improved the lipid profile during pre-diabetes by enhancing LDL clearance, promoting fatty acid oxidation and HDL synthesis. Additionally, the MET + ND and MET + HFHC groups demonstrated similar effects on lipid profiles compared to the GTIN-treated groups. MET has been shown to improve dyslipidaemia by enhancing insulin sensitivity and regulating hepatic lipid metabolism [85,86]. While the MET + ND and MET + HFHC groups were effective in improving lipid profile markers in the PD state, GTIN exhibited more pronounced effects.

Under normal physiological conditions, the regulation of MAP levels involves the integration of neural, hormonal and local mechanisms [87]. NO is a vasodilator that is produced by eNOS and is important in the regulation of vascular tone [88]. However, intermediate hyperglycaemia promotes mitochondrial dysfunction and the formation of advanced glycated end products and triggers inflammatory pathways [89]. These mechanisms contribute to increased oxidative stress, which damages endothelial cells [90,91]. Endothelial dysfunction reduces NO availability, which subsequently increases vascular resistance and MAP levels [90,91]. In this study, MAP levels were significantly higher throughout the treatment period in comparison to the NPD group. This was further accompanied by elevated plasma eNOS levels at week 32 in the PD control group. These results corroborated with previous studies, which have reported similar findings in the PD state [80,92]. The findings may suggest that eNOS levels increased to compensate for elevated MAP levels. However, the GTIN + ND and GTIN + HFHC groups exhibited significantly lower MAP levels at week 28 and week 32 in comparison to the PD control. This was further accompanied by decreased plasma eNOS levels at week 32 of the treatment period. 

Studies have shown that quercetin improves MAP levels by reducing oxidative stress and activating eNOS synthesis [93,94]. Furthermore, previous literature has shown that 15 mg/Kg and 20 mg/Kg GTIN administration ameliorates intermediate hyperglycaemia, reduces oxidative stress and enhances AMP-activated protein kinase (AMPK) activity [41,95]. AMPK activation enhances the NO pathway, which promotes vasodilation [95]. Similarly, in our study, 15 mg/Kg GTIN may have decreased MAP levels by facilitating vasodilation through its effects on intermediate hyperglycaemia and oxidative stress. The plasma eNOS levels in the GTIN + ND and GTIN + HFHC groups may have been attributed to reduced oxidative stress. Additionally, both MET-treated groups exhibited effects on MAP levels that were similar to the GTIN-treated group. This was further accompanied by decreased plasma eNOS levels in the MET + ND group in comparison to the PD control. This may suggest that MET is effective in controlling MAP levels when used in conjunction with dietary intervention.

Adiposity is influenced by caloric intake, energy expenditure and hormonal control [96]. This condition promotes chronic inflammation, which accelerates plaque buildup in the arteries and subsequently increases the risk of CVDs [97]. BMI and WC are useful in monitoring adiposity and visceral fat in the management of pre-diabetes [98,99]. The chronic consumption of high-caloric diets has been shown to contribute to weight gain by stimulating increased lipogenesis, inhibiting lipolysis and interfering with appetite regulation [81]. This further leads to increased BMI and WC observed in PD individuals [100]. In this study, both BMI and WC were significantly higher throughout the treatment period in the PD group in comparison to the NPD group. Previous research has established that chronic high-caloric intake promotes adiposity, which is commonly observed in PD individuals [80,101]. This study’s results support these findings. However, the GTIN + ND and GTIN + HFHC groups showed significantly lower BMI and WC at week 28 and week 32 of the treatment period in comparison to the PD control. In a recent study, it was shown that GTIN administration significantly reduced caloric intake and weight gain by decreasing plasma ghrelin levels [45]. This may account for the reduced BMI and WC circumference observed in the GTIN-treated groups. Interestingly, the MET + ND group showed significantly lower BMI at week 28 in comparison to the PD control. However, both MET-treated groups exhibited similar effects on BMI and WC levels at week 32 as the GTIN-treated group. MET has been shown to reduce BMI and WC by targeting insulin resistance, appetite suppression and hepatic glucose production [102,103]. While both MET-treated groups were effective in reducing BMI and WC by week 32, the GTIN-treated groups had already shown effectiveness by week 28. This may suggest that 15 mg/kg GTIN administration both in the presence and absence of dietary modification exhibits a faster onset of effectiveness than MET.

Under normal conditions, there exists a balance between ROS levels and antioxidants [104]. However, intermediate hyperglycaemia has been shown to promote the accumulation of increased ROS through mitochondrial dysfunction [15,105]. Furthermore, elevated advanced glycated end-products and free fatty acids observed in the PD state trigger inflammation-induced ROS production [106]. The accumulation of ROS contributes to oxidative stress, which triggers lipid peroxidation [107]. MDA is the toxic by-product produced by lipid peroxidation and is responsible for damaging cellular functions [108]. The redox imbalances observed during pre-diabetes may contribute to the development of CVDs [109]. In this study, heart MDA levels were significantly higher in the PD control group compared to the NPD group. This was further accompanied by significantly lower heart SOD and Gpx activity in the PD control. These results may suggest that the reduced antioxidant capacity of heart tissue may have compromised its ability to maintain redox balance, leading to increased lipid peroxidation. However, the GTIN + ND and GTIN + HFHC groups showed significantly lower heart MDA levels in comparison to the PD control. This was further accompanied by increased heart SOD and Gpx activity in both GTIN-treated groups. Previous studies have shown that GTIN exhibits potent antioxidant effects, which have been attributed to its free radical scavenging capability [44,110]. Interestingly, the specific arrangement of hydroxyl and methoxy groups on the flavonoid backbone structure of GTIN (Figure 1) may contribute to its antioxidant effects [41,111]. This study’s results may suggest that 15 mg/kg GTIN administration in both the presence and absence of dietary modification reduces oxidative stress and lipid peroxidation by enhancing the antioxidant enzyme activity of heart tissue. Additionally, both MET-treated groups showed significantly similar effects on heart MDA levels as the GTIN-treated groups. This was further accompanied by increased heart SOD and GPx activity in only the MET + ND group. Previous studies have shown that MET decreases oxidative stress by promoting AMPK activation, improving insulin sensitivity, upregulating antioxidant enzyme expression and improving mitochondrial function [112,113]. This may suggest that MET exhibits beneficial effects on antioxidant enzyme activity when used in conjunction with dietary modification.

Under normal physiological conditions, inflammation is essential in the response to injury and tissue healing [114]. This has to be regulated to prevent chronic inflammation, which promotes tissue damage observed in CVDs [115]. However, during pre-diabetes, oxidative stress has been shown to activate pathways that release pro-inflammatory cytokines such as TNF-α, IL-6 and CRP [15,116]. The inflammatory response recruits and activates immune cells at the site of injury [117]. Activated immune cells produce ROS and cytokines, which further worsen inflammation and tissue damage [118]. In this study, inflammatory markers such as plasma IL-6, TNF-α and CRP levels were significantly higher in the PD control group in comparison to the NPD group. This study’s findings corroborate previous studies that reported the presence of low-grade inflammation in the PD state [116,119]. However, the GTIN + ND and GTIN + HFHC groups showed significantly lower inflammatory marker levels in comparison to the PD control. A previous study showed that GTIN reduces inflammation by targeting ROS accumulation [64]. Therefore, it may be speculated that GTIN reduces low-grade inflammation by targeting oxidative stress due to its potent antioxidant properties [95]. Additionally, both MET-treated groups exhibited effects on these parameters similar to the GTIN-treated groups. Previous studies have shown that MET reduces inflammation by activating the AMPK, which inhibits nuclear factor kappa B (NF-kB) signaling [120,121]. By suppressing NF-κB, MET reduces the production of TNF-α and IL-6, improving cardiovascular function [122]. The anti-inflammatory effects of the conventional anti-diabetic drug MET were comparable to GTIN.

This study highlights the potential of 15 mg/kg GTIN as an effective compound for improving cardiovascular function in the PD state. The results demonstrated that GTIN significantly improved lipid profile markers, reducing plasma TC, TG, LDL and VLDL while increasing HDL in comparison to the NPD group. Additionally, GTIN administration decreased MAP, BMI, WC and plasma eNOS levels. It also reduced heart oxidative stress by lowering MDA levels and enhanced antioxidant activity through increased SOD and Gpx activity. Moreover, GTIN decreased inflammation, as indicated by lower plasma IL-6 and TNF-α levels. Notably, these beneficial effects were observed both in the presence and absence of dietary intervention, highlighting GTIN’s potential role in cardiovascular health management.

## 4. Materials and Methods

### 4.1. Chemicals and Drugs

All chemicals and drugs were of analytical grade and purchased from standard commercial suppliers (Merck Chemicals (PTY) Ltd., Wadeville, Gauteng, South Africa). 

### 4.2. Animals and Housing

This study utilised 36 male Sprague Dawley rats (150–180 g), which were bred and housed in the Biomedical Research Unit (BRU) at the University of KwaZulu-Natal (UKZN), Westville campus. The animals were maintained under standard laboratory conditions, which included a constant temperature of 22 ± 2 °C, a carbon dioxide (CO_2_) content of <5000 p.m., a relative humidity of 55 ± 5% and illumination (12 h light/dark cycle, lights on at 07h00). The noise level was maintained at less than 65 decibels. All animals were allowed access to food and fluids ad libitum. The Animal Research Ethics Committee of the University of KwaZulu-Natal (ethics: AREC/0000495/2022) granted permission for all animal experimentation and protocols. Before exposure to the experimental diets, the rats were allowed to acclimatise to their new environment while consuming standard rat diet and tap water [123]. The care and handling procedures were followed in accordance with the Animal Research Ethics Committee (AREC) of UKZN. 

### 4.3. Induction of Pre-Diabetes 

The animals were randomly assigned into two dietary groups: group A (*n* = 6) and group B (*n* = 30). Experimental pre-diabetes was induced in the animals using a method previously outlined by Luvuno et al. [123]. Group A animals received a standard rat diet with tap water while group B received the HFHC diet supplemented with 15% fructose-enriched water for the induction of pre-diabetes (AVI Products (Pty) Ltd., Waterfall, South Africa). At week 20, the animals were assessed for pre-diabetes based on the American Diabetes Association (ADA) criteria [124]. Animals with a fasting blood glucose concentration of 5.6 to 6.9 mmol/L, 2 h oral glucose tolerance test glucose concentration of 7.8 to 11.0 mmol/L and a glycated haemoglobin concentration of 5.7 to 6.4% were classified as the pre-diabetic group. The animals that received the standard diet were also evaluated at week 20 to confirm normoglycaemia and were classified as the non-pre-diabetic group. 

### 4.4. Experimental Design and Treatment

Following the induction of pre-diabetes, the pre-diabetic group (*n* = 30) was further divided into five sub-groups (Group B to Group F), consisting of six animals each. The pre-diabetic animals either continued the HFHC diet or changed to a normal standard diet (ND) while receiving either 15 mg/kg GTIN or 500 mg/kg MET orally once every third day for 12 weeks. We selected an oral dose of 15 mg/kg GTIN based on previous studies showing its effectiveness and safety within the 10 mg/kg–20 mg/kg range [44,45,125,126]. Similarly, 500 mg/kg MET was chosen due to its proven therapeutic effects at this dose and minimal side effects [127,128,129]. The timing of dosage was chosen to minimise drug accumulation and toxicity [31,130,131]. The non-pre-diabetic control (Group A) animals continued the ND while the pre-diabetic control group (PD, Group B) remained on the HFHC diet and received as the vehicle 3 mL/kg of diluted dimethyl sulphoxide (2 mL DMSO: 19 mL normal saline, p.o.)(Merck Chemicals (PTY) LTD, Wadeville, Germiston, Gauteng, South Africa). Group C (GTIN + ND) animals received GTIN and were changed to the ND diet while Group D (GTIN + HFHC) received GTIN and continued the HFHC diet. The Group E (MET + ND) animals received MET and changed to the ND diet while Group F animals (MET + HFHC) received MET and continued the HFHC diet. Parameters such as body mass index (BMI), waist circumference (WC) and blood pressure were monitored at weeks 20, 24, 28 and 32 as described in the established protocol [31]. The systolic and diastolic pressure of all experimental animals was measured using a non-invasive tail-cuff method and the MAP was calculated using the formula MAP = [(2 × diastolic) + systolic]/3 [31,132].

### 4.5. Blood Collection and Tissue Harvesting

At the end of week 32, all animals were anaesthetised with Isofor (100 mg/kg) (Safeline Pharmaceuticals (Pty) Ltd., Roodeport, South Africa) for 3 min using a gas anaesthetic chamber (BRU, UKZN, Durban, South Africa). Once the animals were unconscious, blood samples were obtained from the animals through cardiac puncture and transferred into individual pre-cooled heparinised containers. The blood samples were centrifuged (Eppendorf AG, Eppendorf centrifuge 5403, Hamburg, Germany) at 4 °C, 503× *g* for 15 min to obtain plasma. Each plasma sample was aspirated into plain sample bottles and stored at −80 °C in a Bio Ultra freezer (Snijers Scientific, Tilburg, The Netherlands) until ready for biochemical analysis. Thereafter, the heart was removed and snap-frozen in liquid nitrogen before storage in a BioUltra freezer (Snijers Scientific, Tilburg, The Netherlands) at −80 °C until biochemical analysis. 

### 4.6. Biochemical Analysis

At the end of week 32, the plasma lipid profile was analysed by measuring TG, TC and HDL according to manufacturer instructions using their respective assay kits (Elabscience Biotechnology Co., Ltd., Houston, TX, USA). The absorbance of the samples for TG and TC was measured at 510 nm, while HDL absorbance was measured at 600 nm using a Spectrostar Nanospectrophotometer (BMG Labtech, Ortenberg, Baden-Württernberg, LGBW, Germany). VLDL and LDL cholesterol levels were calculated using Friedewald’s formula: LDL = TC − HDL − (TG/2.2) and VLDL = 2.2/TG [133]. Heart malondialdehyde [70] levels were assessed following a previously published protocol [134]. Heart SOD and GPx activity as well as plasma eNOS, TNF-α, IL-6 and CRP were measured using their respective ELISA kits (Elabscience Biotechnology Co., Ltd., Houston, TX, USA). 

### 4.7. Statistical Analysis

The statistical data were presented as mean ± SEM. The data were analysed by one-way analysis of variance [116] followed by Tukey–Kramer via GraphPad Prism 5 software. Statistical significance was determined at *p* < 0.05.

## 5. Conclusions

Taken together, GTIN administration both in the presence and absence of dietary modification reduced elevated MAP levels, oxidative stress and inflammation, as well as improved endothelial function and antioxidant enzyme activity in diet-induced pre-diabetes. The results were comparable to those observed with MET administration, indicating that GTIN may serve as a viable alternative for managing cardiovascular complications in the PD state. Nonetheless, further research is necessary to elucidate the mechanisms underlying the observed effects of GTIN administration.

## Figures and Tables

**Figure 1 ijms-25-12105-f001:**
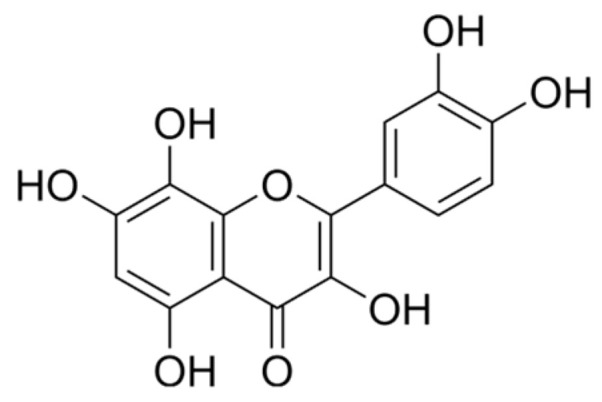
The chemical structure of gossypetin [46].

**Figure 2 ijms-25-12105-f002:**
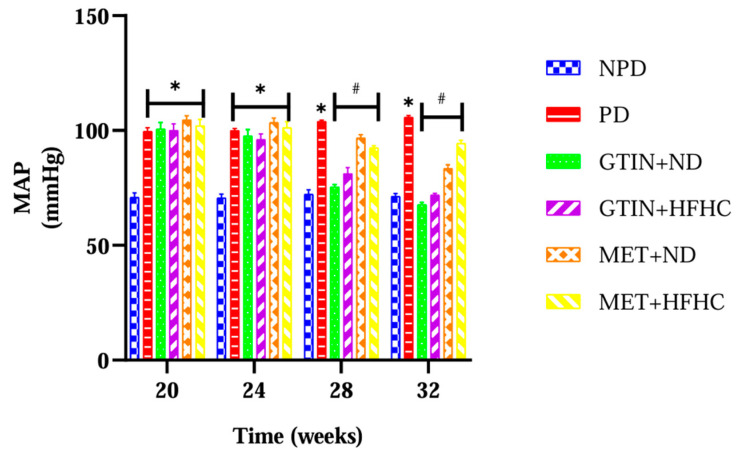
The effects of GTIN on MAP levels in rats in both the presence and absence of dietary modification from week 20 to week 32 of the experimental period (*n* = 6, per group). Values are presented as mean ± SEM. * = *p* < 0.05 denotes comparison with NPD, # = *p* < 0.05 denotes comparison with PD. NPD: non-pre-diabetic control; PD: pre-diabetic control; gossypetin and normal diet (GTIN + ND); gossypetin and high-fat high-carbohydrate diet (GTIN + HFHC); metformin and normal diet (MET + ND); metformin and high-fat high-carbohydrate group (MET + HFHC).

**Figure 3 ijms-25-12105-f003:**
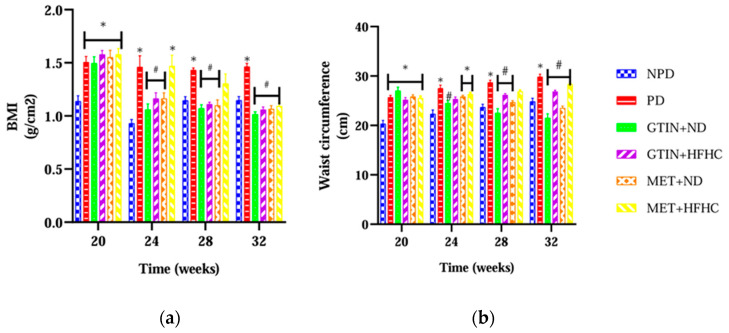
The effects of GTIN on (**a**) BMI and (**b**) WC in both the presence and absence of dietary modification from week 20 to week 32 (*n* = 6, per group). Values are presented as mean ± SEM. * = *p* < 0.05 denotes comparison with NPD; # = *p* < 0.05 denotes comparison with PD. NPD: non-pre-diabetic control; PD: pre-diabetic control; gossypetin and normal diet (GTIN + ND); gossypetin and high-fat high-carbohydrate diet (GTIN + HFHC); metformin and normal diet (MET + ND); metformin and high-fat high-carbohydrate group (MET + HFHC).

**Figure 4 ijms-25-12105-f004:**
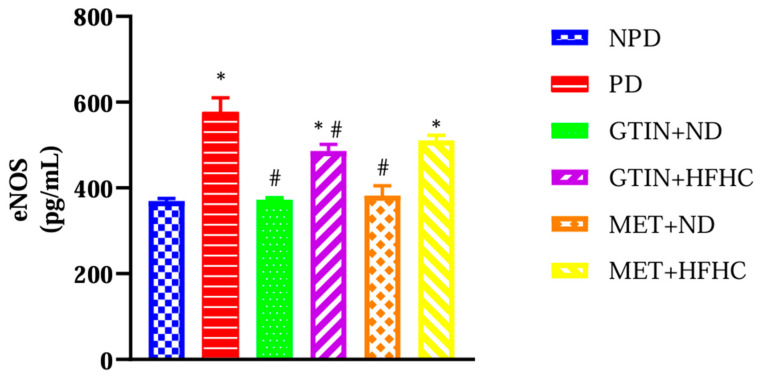
The effects of GTIN on eNOS in both the presence and absence of dietary modification at week 32 of the treatment period (*n* = 6, per group). Values are presented as mean ± SEM. * = *p* < 0.05 denotes comparison with NPD; # = *p* < 0.05 denotes comparison with PD. NPD: non-pre-diabetic control; PD: pre-diabetic control; gossypetin and normal diet (GTIN + ND); gossypetin and high-fat high-carbohydrate diet (GTIN + HFHC); metformin and normal diet (MET + ND); metformin and high-fat high-carbohydrate group (MET + HFHC).

**Table 1 ijms-25-12105-t001:** The effects of GTIN on lipid profile markers in both the presence and absence of dietary modification at week 32 of the treatment period (*n* = 6, per group). Values are presented as mean ± SEM.

Parameters	Experimental Groups					
	NPD	PD	GTIN + ND	GTIN + HFHC	MET + ND	MET + HFHC
**TG** **(mmol/L)**	1.03 ± 0.077	2.27 ± 0.22 *	0.91 ± 0.04 #	1.18 ± 0.050 #	1.05 ± 0.071 #	1.19 ± 0.013 #
**TC** **(mmol/L)**	2.65 ± 0.12	4.36 ± 0.05 *	2.80 ± 0.058 #^	3.08 ± 0.091 #	2.84 ± 0.045 #	3.25 ± 0.076 #
**HDL** **(mmol/L)**	1.38 ± 0.084	0.70 ± 0.12 *	1.52 ± 0.093 #	1.41 ± 0.066 #	1.56 ± 0.11 #	1.22 ± 0.10 #
**LDL** **(mmol/L)**	0.74 ± 0.14	2.79 ± 0.14 *	0.72 ± 0.10 #^	1.06 ± 0.07 #	0.69 ± 0.094 #	1.30 ± 0.066 #
**VLDL** **(mmol/L)**	0.53 ± 0.024	0.87 ± 0.011 *	0.56 ± 0.012 #^	0.62 ± 0.018 #	0.56 ± 0.0065 #	0.64 ± 0.012 #

* = *p* < 0.05 denotes comparison with NPD, # = *p* < 0.05 denotes comparison with PD, ^ = *p* < 0.05 denotes comparison with MET + HFHC. NPD: non-pre-diabetic control; PD: pre-diabetic control; gossypetin and normal diet (GTIN + ND); gossypetin and high-fat high-carbohydrate diet (GTIN + HFHC); metformin and normal diet (MET + ND); metformin and high-fat high-carbohydrate group (MET + HFHC).

**Table 2 ijms-25-12105-t002:** The effects of GTIN on heart oxidative stress, antioxidant and plasma inflammatory markers in both the presence and absence of dietary modification at week 32 of the treatment period (*n* = 6, per group). Values are presented as mean ± SEM.

Parameters	Experimental Groups					
	NPD	PD	GTIN + ND	GTIN + HFHC	MET + ND	MET + HFHC
**MDA** **(nmol/g protein)**	3.95 ± 0.59	7.73 ± 0.18 *	4.42 ± 0.47 #^	5.38 ± 0.23 #	4.97 ± 0.28 #	6.68 ± 0.24 #
**T-SOD Activity** **(U/mgprot)**	19.88 ± 0.89	6.88 ± 0.83 *	17.65 ± 0.59 #^	13.74 ± 0.30 #	16.98 ± 0.73 #	9.80 ± 0.68
**GPx Activity** **(U/mgprot)**	941.00 ± 27.93	415.10 ± 9.69 *	932.60 ± 40.95 #^	837.3 ± 20.84 #	842.10 ± 16.60 #	463.00 ± 49.69
**IL-6** **(pg/mL)**	20.22 ± 3.55	54.98 ± 3.56 *	18.79 ± 1.10 #^	22.60 ± 0.65 #	22.80 ± 0.99 #	25.76 ± 0.89 #
**TNF-α** **(pg/mL)**	14.96 ± 0.31	31.77 ± 0.51 *	13.09 ± 0.31 #^	21.15 ± 3.03 #	13.77 ± 0.61 #	25.83 ± 0.81 #
**hs-CRP** **(pg/mL)**	4.98 ± 0.58	15.91 ± 0.18 *	5.12 ± 0.71 #^	6.58 ± 0.074 #^	6.03 ± 0.18 #	7.74 ± 0.11 #

* = *p* < 0.05 denotes comparison with NPD; # = *p* < 0.05 denotes comparison with PD, ^ = *p* < 0.05 denotes comparison with MET + HFHC. NPD: non-pre-diabetic control; PD: pre-diabetic control; gossypetin and normal diet (GTIN + ND); gossypetin and high-fat high-carbohydrate diet (GTIN + HFHC); metformin and normal diet (MET + ND); metformin and high-fat high-carbohydrate group (MET + HFHC).

## Data Availability

The data presented in this study are available on request from the corresponding author.

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
