# Peer review of "Investigating the Effects of Gossypetin on Cardiovascular Function in Diet-Induced Pre-Diabetic Male Sprague Dawley Rats"

_ijms, 2024, doi:10.3390/ijms252212105_

Round 1
Reviewer 1 Report
Comments and Suggestions for Authors
In the manuscript “Investigating the effects of gossypetin on cardiovascular function in diet-induced pre-diabetic male Sprague-Dawley rats”, Authors evaluate the effectiveness of the natural flavonoid gossypetin in counteracting the metabolic disorders exerted by a pre-diabetic treatment on mice. In particular, authors compare the potential of gossypetin with the well-known metformin in limiting the increase in TAG/cholesterol levels, blood pressure, body mass, oxidative stress and inflammation, processes induced by a diet rich in fatty acids and carbohydrates. Requests for clarification are set out below.
-Figure 1. MAP levels determination. 24 weeks: How is it possible that only the PD group has a significantly different MAP value compared to NPD?
-Figure 2. BMI levels determination. 24 weeks: How is it possible that the GTIN+HFHC, MET+ND and especially GTIN+ND groups are not significant with respect to PD?
-Figure 2 Waist circumference evaluation 24 weeks: regarding the evaluation of the significance of the MET+ND and MET+HFHC groups, a different symbol appears to be indicated from those used to indicate PD vs NDP and treated group vs PD
-Experimental design and treatment. By what criteria were the concentrations of GTIN and MET and the timing of administration chosen?
-Regarding the analysis of plasma lipid profile measurement, the authors should add more information
Reviewer 2 Report
Comments and Suggestions for Authors
All comments, suggestions, and questions are available throughout the manuscript.
